# Multilevel physics informed neural networks (MPINNs)

## Abstract

In this paper we introduce multilevel physics informed neural networks (MPINNs). Inspired by classical multigrid methods for the solution of linear systems arising from the discretization of PDEs, MPINNs are based on the classical correction scheme, which represents the solution of the PDE as the sum of a fine and a coarse term that are optimized in an alternate way. We show that the proposed approach allows us to reproduce in the neural network training the classical acceleration effect observed for classical multigrid methods, thus providing a PINN that shows improved performance compared to the classical ones. Our tests show that, on elliptic and nonlinear equations MPINNs are less sensitive to the choice of the learning rate than classical PINNs, that they can reach the same approximation error as PINNs employing less operations, and in many cases provide a faster and improved decrease of the approximation error.

## 1 Introduction

The approximation of the solution of partial differential equations (PDEs) by artificial neural networks (ANNs) dates back to the 90s (Lagaris et al., 1998), but it is only in these last years that this topic fully emerged and gave rise to an active field of research. ANNs have been used for many different purposes in this field: for the numerical solution of either direct problems (Di Muro & Ferrari, 2008; Raissi et al., 2019; Rudd, 2013; Mishra, 2018; Sirignano & Spiliopoulos, 2018; Luo & Yang, 2020) or of inverse problems (Raissi & Karniadakis, 2018; Raissi et al., 2017b), to reconstruct the equation from given data (Long et al., 2018; Rudy et al., 2017; Schaeffer, 2017), to learn dynamics from incomplete information and physical priors (Ayed et al., 2019; De Bézenac et al., 2019). ODE solvers have even been used for supervised learning problems (Chen et al., 2018). Also the interpretation of successful ANNs like ResNET as ODE discretization schemes is an important direction (Ruthotto & Haber, 2019; Lu et al., 2018), as well as the study of the issues of stability and robustness for different schemes (Haber et al., 2019; Chang et al., 2018).

In this field, the most famous network architecture is the one of PINNs (physics informed neural networks), introduced for the first time in Raissi et al. (2019; 2017a;b). Since their introduction, the PINNs architecture has encountered a growing interest and the good performance observed in practice have been later supported by theoretical results. In Mishra & Molinaro (2020b;a) the authors propose rigorous estimates on the generalization error of PINNs approximating the solution of the direct and inverse problems for PDEs data assimilation problems in terms of the training error and number of training samples, while convergence results for the sequence of approximations to the solution of linear second-order elliptic and parabolic PDEs when the number of data grows are proposed in Shin et al. (2020).

Despite their good performance, the training of such networks may still represent a challenge in case of difficult problems, such as highly nonlinear problems. In this case really large networks may be needed to correctly represent the sought solution, leading to the need of solving a large-scale optimization problem, for which standard training methods may show a slow convergence, and it may be difficult to properly tune the learning rate.

When dealing with linear PDEs, multigrid (MG) methods are by far the most effective methods for the solution of large scale problems (Hackbusch, 1985; Briggs et al., 2000; Trottenberg et al., 2000). The improved performance of MG methods derives from the fact that alternating relaxations among

fine and coarse grids allows us to more efficiently reduce all the components of the error, smooth and oscillatory ones.

The same behaviour can be observed in multilevel optimization for nonlinear problems (Gratton et al., 2008; Groß & Krause, 2009; Kočvara & Mohammed, 2016; Lewis & Nash, 2005; 2013; Nash, 2000; 2014; Wen & Goldfarb, 2009; Calandra et al., 2021), which still exploits representations of the problem at different scales. If in MG methods the residual of the nonlinear equation is transferred from a grid to the other, in such schemes the variables of the optimization problems are the object of the transfer operators. It is evident that the application of such techniques to neural network training is not straightforward, as the transfer operators usually used in MG and multilevel optimization (standard interpolation and restriction operators) requires the quantities that need to be transferred to possess an underlying geometrical structure in order to be effective. In the case of a training problem, the variables subject to optimization are the weights and biases of the network, which do not possess any geometrical structure. To derive a multilevel method for the training problem, it is necessary to design multilevel transfer operators differently and it is not evident how to do that.

Previous contributions have been made in this direction in Calandra et al. (2020). The authors propose a strategy to transfer the weights that is inspired to algebraic multigrid (Ruge & Stüben, 1987; Brandt, 2000), which however requires the knowledge of the Hessian matrix of the loss function, which is impractical for large scale problems. Moreover the proposed approach is designed for one layer neural networks, which limits the range of applications of the approach.

In this work we avoid this problem by following a completely different path: diverging from classical multilevel optimization methods and remaining closer to MG methods, we propose a multilevel PINN approach (MPINN), based on writing the solution of the PDE as a sum of two terms, a fine and a coarse one. Each term is a PINN depending on a different number of parameters and trained on a different training set, which are optimized independently the one from the other, as each PINN has its own weights and training points that are not transferred from a network to the other one. As in classical MG, the method proceeds by alternating relaxations on the two levels, which in this case are epochs of training of each PINN. As it is common in the multilevel literature, the approach is presented in the two-levels framework, but by a recursive scheme it can be naturally extended to the multilevel case. In such case, the solution of the PDE will be written as the sum of a finite number of increasingly coarser networks. Interestingly, this approach allows us to reproduce the acceleration typically observed in classical MG methods, in the context of the training of PINNs.

**Related work** The idea of exploiting multiple scales in learning is not new. We mention for instance Haber et al. (2018); Ke et al. (2017) that propose multiscale methods for convolutional neural networks, which connect low-resolution and high-resolution data, leading to new training strategies that gradually increase the depths of the CNN while re-using parameters for initializations.

A similar idea is employed in Cai & Xu (2019), but in the context of partial differential equations. Multi-scale deep neural networks are introduced, based on specific techniques to convert the learning or approximation of high frequency data to that of a low frequency ones. Still in this context, Fan et al. (2019) introduce neural networks with a novel multiscale structure inspired by hierarchical matrices, which are used to approximate discrete nonlinear maps obtained from discretized nonlinear partial differential equations, such as those arising from nonlinear Schrödinger equations.

In Chung et al. (2017) the authors propose multiscale recurrent networks, that can capture the latent hierarchical structure in temporal sequences by encoding the dependencies with different timescales.

With respect to these methods, our is closer in spirit to classical multigrid and enables at the same time a multilevel structure in the space of the network's parameters and in that of the samples.

An approach similar in spirit to ours, which is derived from another classical technique used for the solution of PDEs (domain decomposition rather than multigrid) is extended physics informed neural networks (XPINNs) (Jagtap & Karniadakis, 2020; Hu et al., 2021). This approach has been theoretically and empirically proved to improve standard PINNs by using multiple sub-networks with different complexities for each subdomain. This is very related to our work: instead of decomposing the domain space, MPINNs are decomposing the solution components for fine and coarse parts.

**Structure of the manuscript** The manuscript is organised as follows. In Section 2 we briefly review the standard PINNs architecture and in Section 3 the standard multigrid method. Section 4

represents the main contribution of this work, where we introduce the multilevel PINNs. In Section 5, we show numerical evidence of the advantage of the MPINNs over classical PINNs. Conclusions and perspectives are presented in Section 6.

## 2   PHYSICS INFORMED NEURAL NETWORKS

PINNs (Raissi et al., 2019) are neural networks that are trained to approximate the solution $u(z)$ of a partial differential equation. To do that, a loss function is defined that is composed of two terms, a term taking into account the physical information and a data-fidelity term. Consider for instance a stationary PDE written as:

$$D(z, u(z)) = f(z), \ z \in \Omega, \quad B(z, u(z)) = f_B(z), \ z \in \partial\Omega,$$

where $\Omega \subseteq \mathbb{R}^d$, $d \geq 1$, is a connected subset, $\partial\Omega$ is the boundary of $\Omega$, $D$ is a differential operator, $B$ is an operator defining the boundary conditions, and $f, f_B : \mathbb{R}^d \to \mathbb{R}$ are given functions. For this problem the loss function reads as:

$$\mathcal{L}(p) = RMSE_{res}(p) + RMSE_{data}(p) \tag{1a}$$

$$RMSE_{res}(p) = \frac{\lambda^r}{N^r} \|D(z, \hat{u}_N(p; z^r)) - f(z^r)\|^2, \tag{1b}$$

$$RMSE_{data}(p) = \frac{\lambda^m}{N^m} \|\hat{u}_N(p; z^m) - u(z^m)\|^2, \tag{1c}$$

where $z^r$ is a vector of $N^r$ sample points in $\Omega$ in which the residual of the PDE is evaluated and $z^m$ is a vector of $N^m$ sample points in $\Omega \cup \partial\Omega$ in which the values of the solution are known; $\lambda^r, \lambda^m$ are weights to balance the two components and $\hat{u}_N(p; z)$ is the sought approximation, which during training is a function of $p$, the set of $N$ weights and biases. Notice that training a PINN does not require the discretization of the operator $D$, as $D(\hat{u}_N(p; z))$ can be directly computed. After training we get the desired approximation $\hat{u}_N(p^*; z)$ to $u(z)$, which depends on the number of parametres used to parametrize the network, $N$, which is a hyperparameter fixed before the training.

## 3   CLASSICAL MULTIGRID METHODS

In this section we briefly review the basic idea behind classical multigrid (MG) methods (Hackbusch, 1985). Consider a linear system arising from the discretization of a PDE:

$$Au = f.$$

Assume to have at disposal some approximation $v$ to the exact solution. There are two important measures of $v$ as an approximation to $u$. One is the error, which is given simply by $e = u - v$, and is however just as inaccessible as the exact solution itself. However, a computable measure of how well $v$ approximates $u$ is the residual, given by $r = f - Av$. The residual is simply the amount by which the approximation $v$ fails to satisfy the original problem $Au = f$. Assuming that the solution of the linear system is unique, we have $r = 0$ if and only if $e = 0$ (however, it may not be true that when $r$ is small in norm, $e$ is also small in norm). Using the definitions of $r$ and $e$, we can derive the so-called *residual equation*, an extremely important relationship between the error and the residual:

$$Ae = r.$$

Given the approximation $v$, it is easy to compute the residual. To improve the approximation $v$, we might solve the residual equation for $e$ and compute a new approximation using the definition of the error $u = v + e$. Usually a relaxation scheme is used to solve the linear systems. Many relaxation schemes possess the so called *smoothing property*, meaning that they are efficient in eliminating the oscillatory modes of the error, while they tend to leave the smooth ones. This is a limiting property of such methods, which is corrected by multigrid methods thanks to this key observation: passing from a fine grid $\Omega_h$ to a coarse grid $\Omega_H$, a mode becomes more oscillatory. The relaxation method will thus efficiently remove the oscillatory components of the error if used on a coarse grid. Mutigrid methods obtain good speed ups by alternating relaxations among fine and coarse grids. The main scheme of MG is the following. Assume to have discretized the problem with two different resolutions, obtaining two subproblems: a fine one $A_h u_h = f_h$ and a coarse one $A_H u_H = f_H$, and assume to have at disposal two linear operators $R$ (restriction) and $P$ (prolongation) to transfer information from a grid to the other one. A V-cycle of MG on two levels follows this scheme:

- Relax $\nu_1$ times on $A_h u_h = f_h$ to obtain an approximation $v_h$

- Compute the residual $r_h = f_h - A v_h$ .

- Project the residual on the coarse level $r_H = R r_h$

- Relax $\nu_2$ times on the residual equation $A_H e_H = r_H$ to obtain $e_H$

- Correct the fine level approximation $v_h = v_h + e_h$, where $e_h = P e_H$.

This procedure is the basis of what is called the *correction scheme*. Having relaxed on the fine grid until convergence deteriorates, we relax on the residual equation on a coarser grid to obtain an approximation to the error itself. We then return to the fine grid to correct the approximation first obtained there. Multiple V-cycles can be performed and the procedure can be extended in a recursive way to more than two levels. It is well-known that MG outperform Gauss Siedel method and already two-grid algorithm can result in tremendous reduction in the iteration count (Mazumder, 2016).

## 4   MULTILEVEL PINNS

Inspired by classical MG, we write the solution of our problem as the sum of two terms:

$$\hat{u}_h(p_h; z_h) + \hat{u}_H(p_H; z_H),$$

where $\hat{u}_h(p_h; z_h)$ corresponds to the "fine" term and is a PINN parametrized by $h$ weights and biases and trained on a fine set $z_h$ of samples, which has enough expressive power to correctly approximate the solution of the PDE, while $\hat{u}_H(p_H; z_H)$ corresponds to the "coarse" term and is a PINN parametrized by $H < h$ parameters and trained on a coarse set of samples $z_H$. The training is inspired to the classical multigrid scheme, which alternates relaxations at coarse and fine level. If we assume that $D$ is linear and that $\lambda^m = 0$, we can recover exactly the classical MG scheme. We define the fine and coarse problems as follows:

$$\min_{p_h} \mathcal{L}_h(p_h) = \frac{1}{N_h^r} \| D(z_h^r, \hat{u}_h(p_h; z_h^r)) - f(z_h^r) \|^2, \tag{2}$$

$$\min_{p_H} \mathcal{L}_H(p_H) = \frac{1}{N_H^r} \| D(z_H^r, \hat{u}_H(p_H; z_H^r)) - r(z_H^r) \|^2. \tag{3}$$

The training in this case follows the following scheme:

- Perform $\nu_1$ epochs on problem 2 to obtain an approximation $\hat{u}_h(p_h, z)$ of $u(z)$

- Compute the residual $r_h(z_h^r) = f(z_h^r) - D(z_h^r, \hat{u}_h)$

- Project the residual on the coarse level $r_H = R(r_h)$

- Perform $\nu_2$ epochs on the residual problem in equation 3 to obtain $\hat{u}_H(p_H, z)$

- Correct the fine level approximation $\hat{u}_h(p_h, z_h) + P(\hat{u}_H(p_H, z_H))$.

The main difference between MG and this approach lies in the definition of the transfer operators $R$ and $P$. In MG they are linear operators, usually standard interpolation ($P_{MG}$) and full-weighting restriction ($R_{MG}$), which are directly applied to the variables of the problem. In this case instead of applying the operators to the variables of the optimization problem $p$, the parameters of the network, we apply them to the underlying geometrical variable $z$, and thus we define:

$$R(\hat{u}_h(p_h, z_h)) := \hat{u}_H(p_H, R_{MG} z_h) \quad P(\hat{u}_H(p_H, z_H)) := \hat{u}_h(p_h, P_{MG} z_H),$$

thus the restriction of a neural network is still a neural network, but with less parameters and that is evaluated on a smaller set of grid points.

In the most general case, we can still use this idea to design a multilevel PINN, by transferring directly the networks from one level to the other rather than the residual, and alternating the minimization at the two levels to correct the approximation at fine level by the information computed at coarse level. We therefore alternate the minimization of two different losses:

$$\mathcal{L}_h(p_h) = RMSE_{res\,h}(p_h) + RMSE_{data\,h}(p_h) \tag{4a}$$

$$RMSE_{res\,h}(p_h) = \frac{\lambda^r}{N_h^r}\|D(z_h^r, \hat{u}_h(p_h; z_h^r) + u_H(z_h^r)) - f(z_h^r)\|^2, \tag{4b}$$

$$RMSE_{data\,h}(p_h) = \frac{\lambda^m}{N_h^m}\|\hat{u}_h(p_h; z_h^m) + u_H(z_h^m) - u(z_h^m)\|^2, \tag{4c}$$

$$\mathcal{L}_H(p_H) = RMSE_{res\,H}(p_H) + RMSE_{data\,H}(p_H) \tag{5a}$$

$$RMSE_{res\,H}(p_H) = \frac{\lambda^r}{N_H^r}\|D(z_H^r, u_h(z_H^r) + \hat{u}_H(p_H; z_H^r)) - f(z_H^r)\|^2, \tag{5b}$$

$$RMSE_{data\,H}(p_H) = \frac{\lambda^m}{N_H^m}\|u_h(z_H^m) + \hat{u}_H(p_H; z_H^m) - u(z_H^m)\|^2, \tag{5c}$$

where $u_i(z) = \hat{u}_i(p_i; z)$ for $i = h, H$ is the approximation computed at the end of each cycle at fine/coarse level. Notice that at coarse level $\hat{u}_H$ is trained on a coarse set of training points $z_H$, but then the found approximation can be evaluated at a larger set of points when it is transferred to the fine level. Notice also that during the fine level training just the fine weights $p_h$ of the fine network $\hat{u}_h$ are optimized, while $u_H(z)$ is a fixed term issued from the previous cycle. Conversely at coarse level $u_h(z)$ is kept fixed. The procedure is described in Algorithm 1.

---

**Algorithm 1** 2-levels training of PINNs

---

1: Input: starting weights $p_h^0, p_H^0$
2: **procedure** $MPINN(p_h^0, p_H^0)$
3:     Set $u_H(z) = \hat{u}_H(p_H^0; z)$.
4:     **for** i=1,2,... **do**
5:         Perform $\nu_1$ epochs for the minimization of equation 4 yielding weights $p_h^*$ and set $u_h(z) = \hat{u}_h(p_h^*, z)$.
6:         Perform $\nu_2$ epochs for the minimization of equation 5 yielding parameters $p_H^*$ and set $u_H(z) = \hat{u}_H(p_H^*, z)$.
7:     **end for**
8: **return** $\hat{u}(p_h^*, z) + \hat{u}(p_H^*, z)$
9: **end procedure**

---

## 5   Numerical results

In this section we validate the performance of the proposed MPINNs in the solution of 1D and 2D elliptical and nonlinear equations. All the codes are implemented in pytorch and the runs are performed using GPUs on Google Colab. The multilevel strategy can be coupled with any optimization scheme to perform the training. Here, we employ two different training strategies: ADAM and LBFGS[1]. For ADAM, the learning rate is experimentally tuned for each problem and it is the same for all networks. More precisely, for the 1D problem we set the starting value to 5e-2, and we decrease it by 0.9 after every 150 iterations; for the 2D problem we set the starting value to 5e-2 and we decrease it by 0.9 every 10 iterations; and for Burger's equation we set the starting value to 1e-3 and decrease it by 0.99 every 500 iterations. For LBFGS we use the Wolfe linesearch routine provided by the method. The training of all the networks is stopped after a maximum number of iteration is reached, this number is large enough to let all the networks reach a steady state. In all the runs we set $\lambda^m = \lambda^r = 1$. We build $z_h^m, z_H^m$ by taking $N_h^m, N_H^m$ equispaced points on the boundary of the considered domain, and we build the points $z_h^r, z_H^r$ as the resulting grid. In all the tests we compare a MPINN composed of a fine network with $h$ neurons per layer and a coarse network with $H$ neurons per layer, the corresponding one level PINN with layers of $h$ neurons (PINN $h$) and a one level PINN (PINN $\tilde{h}$), which have the same total number of parameters as our MPINN. The values of $h$, $H$, $\tilde{h}$ and the number of layers are specified for each test. The performance of the method

---
[1]https://pytorch.org/docs/stable/generated/torch.optim.LBFGS.html

is evaluated by the RMSE between the groundtruth and the computed approximation, evaluated on a grid of unseen points and in terms of *floating point operations* performed during training. This quantity is computed as the number of flops required for the matrix-vector products operations during the training. For the MPINN it takes into account both the operations performed at fine and at coarse level. This gives an idea of the expected gain in the computational time, as the current implementation of MPINNs is not optimized and thus it does not allow us to directly compare the computational time.

## 5.1 1D ELLIPTICAL EQUATION

We consider first a 1D linear problem:

$$u''(z) - u(z) = f(z) \tag{6}$$

with $f(z) = -(\pi^2 + 1)\sin(\pi z) - (\alpha^2\pi^2 + 1)\sin(\alpha\pi z)$ with $\alpha = 3$ on the domain $\Omega = [-1, 1]$ with $u(-1) = u(1) = 0$. The solution of this problem is $\sin(\pi z) + \sin(\alpha\pi z)$. We study the asymptotic behaviour of the MPINNs on Problem 6, that is we consider 1-hidden layer networks with an increasing number of neurons in the hidden layer. In this case $\tilde{h} = h + H$. The values of $h, H$ are specified in Table 1. For each network we report the median and the IQR of the RMSE and training loss computed on a grid of unseen points and of the training loss over 10 runs with random starting weights. We choose $\nu_1 = \nu_2 = 5$ in Algorithm 1. All the networks are trained on a training set of 60 sample points, while the RMSE is evaluated on a finer grid of 100 points.

|  | (h,H) | MPINN | PINN $h$ | PINN $h + H$ |
|---|---|---|---|---|
| RMSE | (50,25) | 1.3e-04, 3.1e-04 | 1.3e-04, 1.2e-04 | 7.0e-04, 4.3e-03 |
| Loss |  | 2.0e-04, 2.1e-04 | 5.3e-04, 3.1e-04 | 2.3e-04, 1.2e-03 |
| RMSE | (200,100) | 2.0e-04, 3.1e-04 | 1.1e-03, 2.9e-03 | 2.0e-03, 2.5e-03 |
| Loss |  | 1.8e-03, 1.2e-03 | 2.9e-04,6.6e-03 | 4.3e-03, 1.9e-02 |
| RMSE | (300,150) | 1.4e-03, 5.2e-03 | 6.1e-03, 9.5e-1 | > 1 |
| Loss |  | 2.0e-02, 2.5e-02 | 1.9e-02, 3.6e-02 | > 1 |

Table 1: Problem 6, $\alpha = 3$. Median and IQR for the RMSE and loss over 10 independent runs with different initial guesses for the weights of the networks.

Results are reported in Table 1. We observe that MPINNs in many runs provide a lower approximation error than standard PINNs. The detail of the evolution of the loss functions and of the RMSE on unseen points during training is reported in Figure 1 for a test with $(h, H) = (50, 25)$. These results show that MPINNs are less sensitive than PINNs to the choice of the learning rate: the rate has been optimized for the case $(h, H) = (50, 25)$ and was used for all the runs. Increasing $h$ MPINNs still perform well with this choice, while the performance of PINNs deteriorates and a new tuning of the learning rate becomes necessary, especially for the largest network. This is a particularly desirable feature in a network, as the tuning of the learning rate may be tedious.

**Remark 5.1** *In the plots we have reported the losses for the different networks all in the same plot, we stress however that they are not directly comparable, as they are defined in different ways for the PINNs and MPINNs (cf. equation 1 and equation 9). The aim of the plots is to assess the development of the training and the choice of the learning rate.*

However, even with LBFGS, which uses a Wolfe linesearch and therefore the learning rate is automatically tuned to ensure the decrease of the loss function, MPINNs achieve a better approximation error than PINNs for this test problem, cf. Figure 2. The advantage becomes even more important in the case of deeper networks: in Figure 2 left we use a 1-hidden layer network, while in Figure 2 right we use a 9-hidden layer network, with $h = 200, H = 100$, all trained by LBFGS.

Besides these favorable features of MPINNs, another important advantage is the fact that, even when the same error level of the PINNs is reached, this is achieved in less iterations. MPINNs provide in most runs a quicker decrease of the error and thus a solution of comparable accuracy to that of PINNs in a reduced number of both iterations and floating point operations, cf. Figure 3.

Multigrid techniques are particularly effective when solving PDEs whose solution has different modes. In Table 2 we thus study the behaviour of the methods for larger values of $\alpha$. In this case the

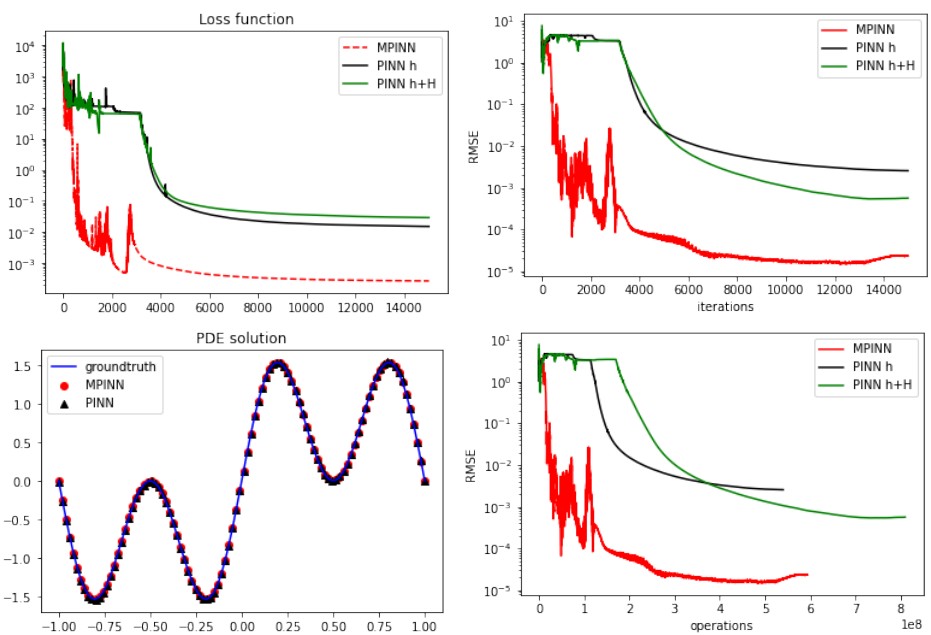

Figure 1: Problem 6, $\alpha = 3$. Comparison of MPINN, PINN $h$, PINN $h + H$ for $h = 50, H = 25$, ADAM. Up: loss function (left) and RMSE (right) values along the iterations. Bottom left: comparison of true solution of the PDE and approximations computed by PINN 200 and MPINN. Bottom right: evolution of the RMSE as a function of the number of floating point operations required for the training of the networks (total number of operations for matrix-vector products).

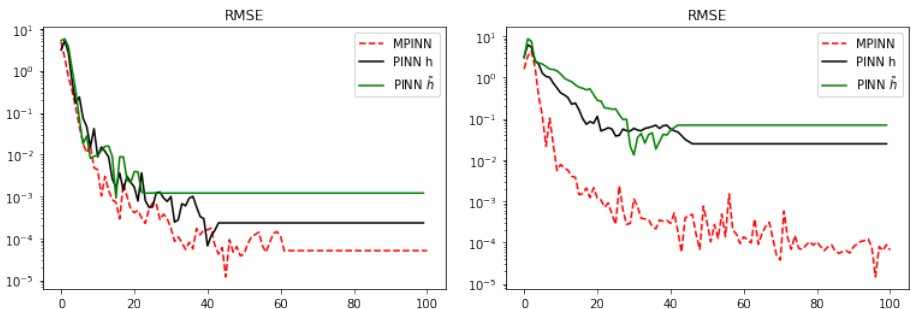

Figure 2: Problem 6, $\alpha = 3$. Comparison of MPINN, PINN $h$, PINN $h + H$ for $h = 200, H = 100$, LBFGS. RMSE over unseen points along the iterations. Left: 1-layer network. Right: 9-layer network.

solution is highly nonlinear. We use networks trained over 400 fine points and 200 coarse points, with $h = 50$, $H = 25$, $\tilde{h} = 54$, 5 layers and we use LBFGS. We can observe that the multilevel strategy is indeed effective also in this context. MPINNs provide a RMSE one order of magnitude lower than classic PINNs and with lower IQR.

| $\alpha$ | MPINN | PINN $h$ | PINN $\tilde{h}$ |
|---|---|---|---|
| 8 | 3.0e-3, 3.0e-3 | 1.5e-2, 2.2e-2 | 1.7e-2, 3.0e-2 |
| 10 | 1.0e-2, 3.1e-2 | 1.3e-1, 2.8e-1 | 4.0e-2, 1.8e-1 |
| 12 | 3.0e-2, 1.0e-1 | 1.0e-1, 3.5 | 1.7e-1, 1.4 |

Table 2: Problem 6, varying $\alpha$, $h = 50$, $H = 25$, $\tilde{h} = 54$, $N_h = 400$, $N_H = 200$, 5 layers. Median and IQR for the RMSE error over 10 independent runs with different initial guesses for the weights of the networks.

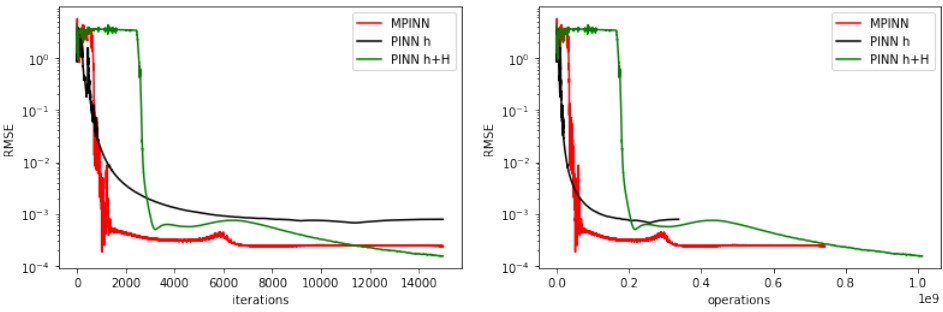

Figure 3: Problem 6, $\alpha = 3$. Comparison of MPINN, PINN $h$, PINN $h + H$ for $h = 50, H = 25$, ADAM. Left: RMSE over unseen points along the iterations. Right: Number of operations along the iterations.

In the Appendix we discuss the choice of the hyperparameters specific to the multilevel strategy, cf. section B.

## 5.2  2D NONLINEAR EQUATION

We consider now a 2D nonlinear problem:

$$-\Delta u + \alpha e^u = f \text{ in } \Omega = [0,1] \times [0,1] \tag{7}$$
$$u(0,y) = 0 \,, u(1,y) = 0, \ \forall y \in [0,1]$$
$$u(x,0) = 0 \,, u(x,1) = 0, \ \forall x \in [0,1]$$

where $f$ has been chosen in order to have the following function as exact solution:

$$u(x,y) = [\sin(\pi x) + \sin(3\pi x)][\sin(\pi y) + \sin(3\pi y)]$$

depicted in Figure 6. We consider again 1-hidden-layer networks and we compare MPINN, PINN $h$, and PINN $h + H$, all trained by ADAM, we train the fine network on a grid of 484 points and the coarse one on a coarser grid of 400 points. The results of the tests are reported in Table 3. More results are available in the Appendix, for different choices of $N_H$. Results are similar to the 1D case, we can again see that MPINNs are less sensitive than classic PINNs to the choice of the learning rate.

|        | $(h, H)$  | MPINN$(h, H)$ | PINN $h$ | PINN $h + H$ |
|--------|-----------|---------------|----------|--------------|
| RMSE   | (30,15)   | 5.5e-2, 1.4e-2 | 8.1e-2, 4.0e-2 | 5.0e-2, 1.0e-2 |
| Loss   |           | 1.2e-1, 0.6e-1 | 4.4e-1, 2.0e-1 | 1.3e-1, 0.5e-1 |
| RMSE   | (100,50)  | 3.0e-2, 1.3e-2 | 3.8e-2, 2.2e-2 | 2.8e-2, 3.9e-1 |
| Loss   |           | 2.5e-2, 1.4e-2 | 7.0e-2, 1.4e-2 | 4.6e-2, 5.0e-1 |
| RMSE   | (200,100) | 3.0e-2, 3.0e-2 (8/10) | 3.0e-2, 3.7e-2 (8/10) | > 1 |
| Loss   |           | 3.0e-2, 6.0e-2 | 5.2.e-2, 1.1e-1 | > 1 |
| RMSE   | (300,150) | 3.0e-2, 3.0e-2 (7/10) | > 1 | > 1 |
| Loss   |           | 5.4e-2, 4.1e-2 | > 1 | > 1 |

Table 3: Problem 7, $\alpha = 0.1$. Median and IQR of the RMSE and loss over 10 independent runs with different initial guesses for the weights of the networks. When numbers in parenthesis are reported it means that not all the 10 runs were successful, i.e. that the achieved RMSE was larger than 1.

## 5.3  BURGER'S EQUATION

Burgers' equation is a fundamental partial differential equation occurring in various areas of applied mathematics. For a given field $u(x,t)$ and diffusion coefficient $\nu$, the general form of Burgers' equation in one space dimension is the dissipative system:

$$\frac{\partial u}{\partial t} + u \frac{\partial u}{\partial x} = \nu \frac{\partial^2 u}{\partial x^2}. \tag{8}$$

We have conducted preliminary tests on this equation, cf. Table 4. As in the other cases, MPINN can reach solution approximations of accuracy comparable to those obtained by classical PINNs, but performing far less operations. We consider here PINNs with 3 hidden layers. We compare a MPINN composed of a PINN with 20 neurons and a PINN with 10 neurons in the hidden layers, and two standard PINNs with 40 and 30 neurons in the hidden layers. The PINNs are trained on a training set of 40 spatial points and 120 temporal points. The MPINN uses 80 time points on the coarse grid. The decrease of the RMSE as a function of the number of operations is depicted in Figure 4 (left). On the right, where we report the RMSE along some iterations, we can observe the effect of the coarse iterations in details. We distinguish in red the RMSE obtained after a set of 5 coarse iterations, and in blue the one obtained after a set of 5 fine iterations. We can see that the coarse iterations help to make the RMSE decrease faster.

|  | MPINN | PINN 40 | PINN 30 |
|---|---|---|---|
| RMSE | 1.3e-1,0.4e-1 | 1.8e-1, 0.4e-1 | 1.7e-1, 1.5e-2 |
| Loss | 2.6e-2, 0.6e-2 | 2.8e-2, 1.1e-1 | 8.5e-3, 7.7e-2 |
| Operations | 1 | 6.1 | 3.5 |

Table 4: Problem 8. Number of operations and median and IQR of the RMSE error and loss function over 10 independent runs with different initial guesses for the weights of the networks.

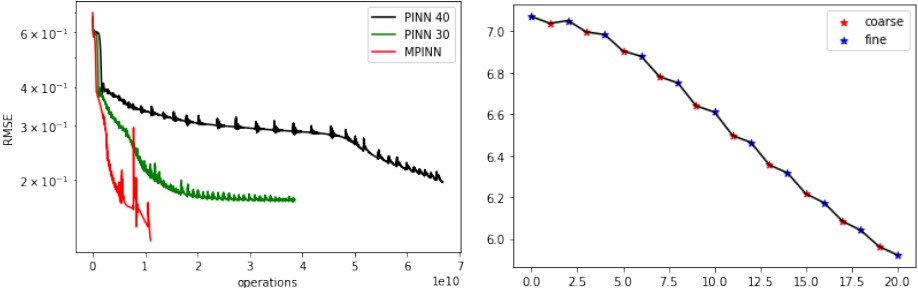

Figure 4: Problem 8. Left: RMSE as a function of the number of floating point operations. Right: Focus on a subset of V-cycles. The blue stars mark the RMSE at the end of a fine pass in the V-cycle, while the red stars mark the RMSE at the end of a coarse pass in the V-cycle.

## 6   CONCLUSIONS AND FUTURE PERSPECTIVES

We have presented MPINNs, a family of multilevel PINNs, whose training is based on the standard correction scheme used by multigrid methods, state-of-the-art methods for the solution of linear elliptical equations. Miming the correction scheme we represent the solution of the PDE as the sum of a fine and a coarse networks, which are optimized in an alternate way. We show the results of tests performed on1D and 2D problems, with elliptical and nonlinear differential operators. The preliminary results show good performance of the MPINNs compared to standard PINNs both in terms of accuracy reached in the solution and in terms of operations in the training. MPINNs also appears to be much less sensitive than PINNs to the choice of the learning rate. As meaningful future research perspectives we target two directions: a deeper numerical investigation of MPINNs, in particular the use of a deeper recursion scheme, as described in Appendix A, and the development of a sound convergence theory for our method.

**Broader impact**   The broader impact of deep learning methods for solving PDEs has been detailed in Um et al. (2021).

**Reproducibility**   We have included our code in the supplementary material for reproducibility.

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

## A    APPENDIX - DEEPER MULTILEVEL SCHEMES

It is well-known that multigrid techniques outperform standard Gauss Siedel method and already two-grid algorithm can result in tremendous reduction in the iteration count (Mazumder, 2016). We have shown that very good results can be obtained by two-levels MPINNs as well.

However, as for standard multigrid, the procedure can be extended to the case of more than two levels in recursive form. This is described in Algorithm 2 where we assume to use $L$ levels. We write $u(z) = \sum_{l=1}^{L} \hat{u}_l(p_l, z)$, with $\hat{u}_l(p_l, z)$ for $1 \leq l \leq L$ a PINN depending on a set $p_l$ of $l$ parameters and trained on training sets $z_l$ (in the algorithm we assume for simplicity that $z_l^m = z_l^r := z_l$), and we assume that for both the parameters set and the samples sets the dimension increases with $l$ (1 is

the minimum level). At each level $l$, $\nu_l$ training iterations are performed on the network $l$ with the following loss:

$$\mathcal{L}_l(p_l) = RMSE_{resl}(p_l) + RMSE_{datal}(p_l) \tag{9a}$$

$$RMSE_{resl}(p_l) = \frac{\lambda^r}{N_l} \|D(z_l, \hat{u}_l(p_l; z_l) + \sum_{i=1, i \neq l}^{L} u_i(z_l)) - f(z_l)\|^2, \tag{9b}$$

$$RMSE_{datah}(p_l) = \frac{\lambda^m}{N_l} \|\hat{u}_l(p_l; z_l) + \sum_{i=1, i \neq l}^{L} u_i(z_l) - u(z_l)\|^2. \tag{9c}$$

---

**Algorithm 2** V-cycle of MPINNs with L levels (recursive form)

---

1: Input: level number $l$ ($1 \leq l \leq L$, 1 is the minimum level), starting values of the networks parameters $p_i$, $i = 1, \ldots, L$
2: **procedure** $p_l, u_l(z) = MPINN(l, p_i \ i = 1, \ldots, L)$
3:     **for** $i = 1, \ldots, L, i \neq l$ **do** set $u_i(z) = \hat{u}_i(p_i, z)$
4:     **end for**
5:     Iterate $\nu_l$ times to minimize equation 9 to obtain $p_l^*$
6:     Set $p_l = p_l^*$
7:     **if** $l = 1$ **then return** $p_1, u_1(z)$
8:     **else**
9:         $p_{l-1}, u_{l-1}(z) = MPINN(l - 1, p_i \ i = 1, \ldots, L)$.
10:        Set $\hat{u}(z) = \hat{u}_l(p_l; z) + \sum_{i=1, i \neq l}^{L} u_i(z)$.
11:    **end if**
12: **end procedure**

---

The numerical investigation of this framework is left as a meaningful future research perspective.

## B   APPENDIX - HYPERPARAMETERS SETTING

.

In this section we discuss the influence of the choice of the hyperparameters (those specific to the multilevel method) on the performance of MPINNs. We discuss in particular the choice of $N_H$ and $H$ with respect to $N_h$ and $h$, respectively.

In the runs in section 5.1 we selected the same number of points in the fine and in the coarse grids. In Table 5 we study the influence of the choice of $N_H := N_H^r$ as compared to $N_h := N_h^r$ in problems 4, 5. These results are also reported in Figure 5, left. The number of operations is computed as the ratio of the number of operations for $H$ and the number of operations for $H = 150$, the maximum value used. We choose $(h, H) = (200, 100)$. This is of course not a comprehensive study, but we can see that decreasing the number of points the accuracy achieved by the MPINN also decreases, even if the behaviour is not strictly regular. If too few points are used the MPINN does not converge.

| $N_H$ | 25 | 50 | 60 | 70 | 100 | 150 |
|---|---|---|---|---|---|---|
| RMSE (median) | 2.3 | 8.0e-4 | 9.4e-4 | 2.8e-4 | 4.5e-4 | 2.3e-4 |
| RMSE (IQR) | 2.3 | 1.9e-3 | 1.6e-3 | 3.1e-3 | 3.1e-4 | 3.0e-4 |
| Operations | 0.85 | 0.88 | 0.89 | 0.84 | 0.94 | 1 |

Table 5: Problem 6, $\alpha = 3$, MPINN with $(h, H) = (200, 100)$, $N_h = 150$ and different choices of $N_H := N_H^r$. Median and IQR (10 independent runs) of the RMSE error on unseen points and ratio of the number of operations performed with respect to the choice $N_H = 150$.

In the runs in section 5.1 we have chosen $H = h/2$, in the spirit of classical multigrid methods, but larger or smaller networks can be employed. In Table 6 we study the influence of the choice of $H$ with respect to a fixed $h$ in problems 4, 5, these values are also reported in Figure 5, right.

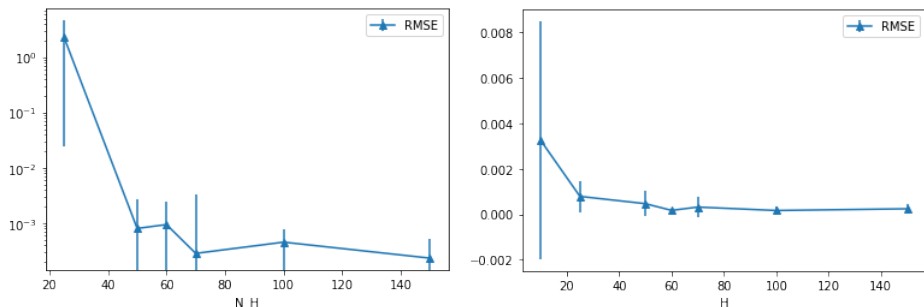

Figure 5: Problem 6, $\alpha = 3$. MPINN with $h = 200$, $N_h = N_H = 150$, varying $H$, ADAM. Median and IQR (10 independent runs) of the RMSE error on unseen points varying parameter $H$ and ratio of the number of operations performed with respect to the choice $H = 150$.

We choose $h = 200$. This is again not a comprehensive study, but we can see that decreasing the number of neurons the accuracy achieved by the MPINN also decreases and the IQR of the results increases. However, the MPINN converges even with few neurons.

The choice of these parameter should thus been based on a compromise between desired accuracy and available training time.

| $H$ | 10 | 25 | 50 | 60 | 70 | 100 | 150 |
|---|---|---|---|---|---|---|---|
| RMSE (median) | 3.2e-3 | 7.8e-4 | 4.6e-4 | 1.7e-4 | 3.1e-4 | 1.6e-4 | 2.4e-4 |
| RMSE (IQR) | 5.2e-3 | 6.8e-4 | 5.5e-4 | 9.0e-5 | 4.5e-4 | 1.9e-04 | 2.2e-4 |
| Operations | 0.88 | 0.89 | 0.91 | 0.93 | 0.96 | 0.98 | 1 |

Table 6: Problem 6, $\alpha = 3$, MPINN with $h = 200$, $N_h = N_H = 150$, varying $H$, ADAM. Median and IQR of the RMSE error over 10 independent runs with different initial guesses for the weights of the networks.

| | MPINN$(h, H)$ | | PINN$(h)$ | | PINN$(\tilde{h})$ | |
|---|---|---|---|---|---|---|
| $n_H$ | RMSE | OP | RMSE | OP | RMSE | OP |
| 5 | 1.6e+00,6.4e-01 | 3.8e+11 | 5.2e-02,1.1e-02 | 3.8e+11 | 4.6e-02,3.8e-03 | 8.6e+11 |
| 15 | 4.7e-02,2.7e-02 | 3.9e+11 | 5.2e-02,1.1e-02 | 3.8e+11 | 4.6e-02,3.8e-03 | 8.6e+11 |
| 25 | 4.0e-02,8.0e-03 | 4.1e+11 | 5.2e-02,1.1e-02 | 3.8e+11 | 4.6e-02,3.8e-03 | 8.6e+11 |
| 35 | 3.9e-02,7.9e-03 | 4.3e+11 | 5.2e-02,1.1e-02 | 3.8e+11 | 4.6e-02,3.8e-03 | 8.6e+11 |

Table 7: Problem 7, $\alpha = 0.1$. Median and IQR of the RMSE error over 10 independent runs with different initial guesses for the weights of the networks and number of operations (OP).

We consider now in Table 7 the effect of $N_H$ in the 2D problem in section 5.2. We consider 3-layers networks with $h = 50$, $H = 25$, $\tilde{h} = 54$ and fixed learning rate of 1.5e-3 for all networks. We observe that for a too small number of points, the coarse network does not converge and does not allow the convergence of the method. On the other hand, with a coarse grid of reasonable size, our method obtains results comparable or even better than the classical PINNs of size $h + H$ while keeping a number of operations (OP) comparable to PINNs of size $h$. The sampling of the coarse grid is, as in the 1D case, a hyperparameter in its own right that must be balanced in order to obtain a coarse network that is good enough to improve convergence while limiting its impact on the computational cost.

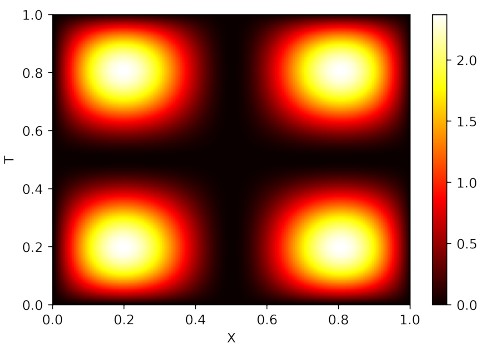

Figure 6: Solution to problem 7

