# OpenReview forum: "Multilevel physics informed neural networks (MPINNs)"
_ICLR.cc/2022/Conference — ICLR 2022 Submitted_

### Official Review · Reviewer_EmLF · 2021-10-25

**Correctness:** 3
**Technical Novelty And Significance:** 3
**Empirical Novelty And Significance:** 3
**Recommendation:** 5
**Confidence:** 4

**Main Review:**

This paper is not well written in English. But since this MPINNs show interesting point that it can run faster and provide smaller approximation error for some PDE equations than using original PINNs, I recommend this paper to be published after polishing the language and also addressing all my below comments.

My comments are as follows:

1.	In the third line on page 5, the formula for R and P has extra ‘)’ at the end. On the same page after the Equation (6c) line, it should be ‘at the end of each cycle’ instead of ‘at the and of each cycle’. In this paper, there are too many ‘really’ and ‘then’.
2.	In this paper, all the experiments are done with h=2*H. Why do you choose 2 here? How does this number matter for the MPINNs training? Some explanations about how to choose this number should be added.
3.	On page 6, the authors state “Indeed, when the number of parameters increase the peculiar structure of the MPINN allows to make the training process easier and as a result the network reaches a better approximation.” The authors should explain why MPINN works better instead of just stating the observation.
4.	On page 7, on the last two lines the authors say “In 2D problems the effect of the size of the training set on the cost is far more important than in the 1D case, therefore we train the fine network on a grid of 484 points and the coarse one on a coarser grid of 400 points”. What is the rule of thumb to choose the grid size?
5.	On page 8, the Figure 3 uses h=10 case while in Table 3 it starts with h=30. Since Figure 3 and Table 3 are talking about the same experiment “2D NONLINEAR EQUATION”, why do you choose different h for table and plot?


**Summary Of The Paper:**

The manuscript entitled “MULTILEVEL PHYSICS INFORMED NEURAL NETWORKS
(MPINNS)” introduces multilevel physics informed neural networks (MPINNs), which is inspired by classical multigrid methods for the solution of linear systems arising from the discretization of PDEs. The authors showed MPINNs provide a faster and smaller approximation error in the case both of elliptic and nonlinear equations and a more robust asymptotic behavior.


**Summary Of The Review:**

This paper proposed the multilevel physics informed neural networks (MPINNs), which is inspired by classical multigrid methods for the solution of linear systems arising from the discretization of PDEs. It showed some interesting results, but those results are lacking of explanation in depth. The authors are mostly just stating the observations, which is not enough to meet the standards of a good paper.

In summary, I would not recommend its publication in ICLR until all my comments are addressed.

---

> ### Author Response · Authors · 2021-11-22
> **Answer to reviewer 4**
>
> We are really grateful to all the referees for their comments which helped us a lot to revise our paper.
>
> > In the third line on page 5, the formula for R and P has extra ‘)’ at the end. On the same page after the Equation (6c) line, it should be ‘at the end of each cycle’ instead of ‘at the and of each cycle’. In this paper, there are too many ‘really’ and ‘then’.
>
> We thank the referee for pointing this typos out, they have been corrected and we tried to polish the language at best we can.
>
> > In this paper, all the experiments are done with h=2*H. Why do you choose 2 here? How does this number matter for the MPINNs training? Some explanations about how to choose this number should be added. On page 7, on the last two lines the authors say “In 2D problems the effect of the size of the training set on the cost is far more important than in the 1D case, therefore we train the fine network on a grid of 484 points and the coarse one on a coarser grid of 400 points”. What is the rule of thumb to choose the grid size?
>
> We have added a study on the hyperparameters tuning for MPINNs in the appendix.
>
> > On page 8, the Figure 3 uses h=10 case while in Table 3 it starts with h=30. Since Figure 3 and Table 3 are talking about the same experiment “2D NONLINEAR EQUATION”, why do you choose different h for table and plot?
>
> The figure was chosen to show the behaviour of MPINNs when a very small architecture is selected. In the current version the figure has been removed in order to fit in the prescribed number of pages.

---

### Official Review · Reviewer_Ugje · 2021-10-26

**Correctness:** 3
**Technical Novelty And Significance:** 4
**Empirical Novelty And Significance:** 1
**Recommendation:** 8
**Confidence:** 4

**Main Review:**

The method of MPINNs makes sense and seems to be novel. Although it is motivated via the classical multigrid method with the operators P and R, the final method is a very simple alternate optimization with two sub-networks (for two levels) where one sub-networks minimizes the loss for ‘fine’ points, and another minimizes the loss for ‘coarse’ points. Because the solution of PDEs can contain fast/fine components as well as coarse/slow components, the proposed method makes sense and I would expect that this works well for such PDEs. In this regard, I really like the paper and recommend acceptance subject to very minor revisions for several concerns that I explain below.

While the idea and method are great, the demonstration of the method can be improved a lot. The present experimental results are somewhat inconclusive because of the experimental settings and the definitions of the plotted values.  For the experimental setting, the authors mention that the learning rates are experimentally tuned, and the best-decreasing strategy is adopted for each network. This can be problematic without presenting more details, as the learning rates can be the reason for the reported superior performance of MPINNs. I would recommend reporting the results with various learning rates or fixing the learning rates a priori by directly using the learning rates of previous work of PINNs. In the current form, I cannot tell if we are overfitting the data and problems with the tuning of learning rates of MPINNs. At least, the authors should report the learning rates for each case.

The idea is very related to extended physics informed neural networks (XPINNs), which was theoretically studied by Hu et al [1]:

[1] Hu et al. When Do Extended Physics-Informed Neural Networks (XPINNs) Improve Generalization?

The previous work [1] has theoretically and empirically shown that XPINNs can improve standard PINNs by using sub-networks, each for different subdomains with different complexities of the sub-solutions. This is very related to MPINNs. Instead of decomposing the domain space, MPINNs are decomposing the solution components for fine and coarse parts. Therefore, the authors should cite the previous work [1] to mention this relationship. Moreover, unlike the previous work on XPINNs [1], the present paper does not provide any theoretical results, which should be also mentioned as a limitation of MPINNs, when compared to XPINNs, at this moment. In sum, the relationship and the relative limitation should be made clear.

For the definitions of the plotted values, the authors should use a metric that can be comparable for all methods. Instead of these loss and objective used for training, the authors can plot the value of errors for the solution of PDEs, by using a set of new unseen data points.

What are the precise definitions of RMSE and loss used in the tables and figures?

I am not entirely convinced by the results where the performance degrades as we increase h and H. Do you observe the same phenomena with BFGS instead of ADAM? It is possible that this can be caused by bad training practices by the authors. I would suggest plotting the training loss over epoch for different h and H as well to see what is going on there.

The paper makes several random claims without any evidence or support. I recommend deleting those claims from the paper. For example, the authors say that it shows improved performance compared to the state-of-the-art, but no state-of-the-art performance is cited or compared against in this paper. After Table 1, it reads “We observe that MPINNs clearly outperform classical PINNs.”, but it is not clear at all because of the uncertain learning rates and the stopping criteria. The following claim is also not supported by any evidence in this paper and it is an opinion of the authors instead of a finding in a scientific paper: “It is clear that structuring the parameters in a clever way is more beneficial than just augmenting their number, to gain both computational time and expressivity.”

In table 2, F is not used and the following sentence should be deleted in its caption: ”F” means a failure in all the runs.

In table 4, is “0.04.6e-2” typo? On page 5, “the and of” should be replaced by “the end of”.



**Summary Of The Paper:**

This paper proposes multilevel physics informed neural networks (MPINNs). When compared to standard PINNs, MPINN uses additional networks for different levels with fine and coarse terms in the view of two-levels. By reclusive definition, this can be generalized to any multi-levels. The method is motivated via classical multigrid method, and is empirically studied with 3 simple ODEs/PDEs.

**Summary Of The Review:**

While there are several concerns and weak evidence to support the main claims, the idea is very novel and significant. I infer from the method and its idea that the additional evidence to support the main claims will come by sooner or later, either by this paper or other authors. There are so many incremental papers without any originality that have been published just because of large-scale experiments for demonstrations. In contrast, this paper has weak demonstrations but a good original idea. Whereas both types of papers have their own merits, I think that the latter type contributes to our community, science, and academia in a longer-term. Accordingly, I recommend this paper for acceptance, subject to minor revisions.

---

> ### Author Response · Authors · 2021-11-22
> **Answer to reviewer 3**
>
> We are really grateful to all the referees for their comments which helped us a lot to revise our paper.
>
> >In the current form, I cannot tell if we are overfitting to the data and problems with the tuning of learning rates of MPINNs. At least, the authors should report the learning rates for each case.
>
>  In the tests we have actually chosen the same learning rate (LR) for the three PINNs and we adjusted it manually for each problem. This is now stated at the beginning of Section 5. We believe the enhanced behaviour of MPINNs does not depend on the choice of the LR, as when we use BFGS the LR is chosen by the linesearch routine implemented in the pytorch LBFGS method. We agree that the learning rate we chose may not be the best possible one, but our tests highlight also that MPINNs are less sensible to the choice of the learning rate as compared to PINNs, for instance when we increase the size of the network, cf. Table 1, MPINN still performs well with the chosen LR, while PINNh requires a new tuning, as its limited performance is actually caused by the optimizer, as shown by the use of BFGS on the same problems, cf. Figure 2. Finally, we point out that the main advantage of MPINNs is not to provide a lower RMSE (which sometimes is also the case) but rather to achieve the same error level of classical PINNs with less operations, cf. Figure 1d and Figure 3.
>
> > The idea is very related to extended physics informed neural networks (XPINNs), which was theoretically studied by Hu et al [1]:
>
> [1] Hu et al. When Do Extended Physics-Informed Neural Networks (XPINNs) Improve Generalization?
>
> Thanks a lot for pointing this reference out, we have discussed the positioning of our work with respect to XPINNs in the introduction and future perspective for theoretical analysis in the conclusions.
>
> > For the definitions of the plotted values, the authors should use a metric that can be comparable for all methods. Instead of these loss and objective used for training, the authors can plot the value of errors for the solution of PDEs, by using a set of new unseen data points.
>
> What are the precise definitions of RMSE and loss used in the tables and figures?
>
> The plotted values in the figures are the loss function being minimized, to assess the choice of the LR, and the RMSE (as in the tables) is actually computed on a set of unseen points.
>
> > I am not entirely convinced by the results where the performance degrades as we increase h and H. Do you observe the same phenomena with BFGS instead of ADAM? It is possible that this can be caused by bad training practice by the authors. I would suggest to plot the training loss over epoch for different h and H as well to see what is going on there.
>
> As said above, the limited performance is indeed caused by the optimizer, as shown by the use of BFGS on the same problems, cf. Figure 2. Thanks for your suggestion. This helped us to show that MPINNs are less sensitive than PINNs to the choice of the LR.
>
> > The paper makes several random claims without any evidence or support.
>
>  We thank the referee for pointing this out, the sentences have been rewritten.
>
> > In table 2, F is not used and the following sentence should be deleted in its caption: ”F” means a failure in all the runs. In table 4, is “0.04.6e-2” typo? On page 5, “the and of” should be replaced by “the end of”.
>
>   We thank the referee for pointing this out, all the typos have been corrected.

---

> > ### Comment · Reviewer_Ugje · 2021-11-26
> > **My concern is all addressed**
> >
> > My concern is all addressed. Thank you!

---

### Official Review · Reviewer_9RP9 · 2021-11-01

**Correctness:** 3
**Technical Novelty And Significance:** 3
**Empirical Novelty And Significance:** 3
**Recommendation:** 5
**Confidence:** 4

**Main Review:**

Using multigrid methods as inspiration for training and designing PINNs is neat. Splitting the solution into two terms is interesting, because this scheme presents the model with an easy and a hard task (learning fine-scale structure is typically more challenging than learning coarse-scale structures). The proposed scheme can also be viewed as some form of data augmentation and it would be interesting to discuss this aspect in more detail. Further, it would be interesting to discuss whether there is some connection between the proposed multilevel learning scheme and Curriculum Learning.

Overall, the presentation of the materials is very clear and the paper reads nicely. However, a major shortcoming of the paper are experiments.

I would like to see the following questions addressed:

* How do you construct $z_h$ and $z_H$? How do you determine whether $z_H$ is not expressive enough? Is this an automatic process or does this require manual tuning? How is the performance affected if $z_H$ is varied from less coarse to more coarse?

* How does the proposed multilevel PINN compare to other state-of-the art PINNs or to classical scientific computing methods for the problems under consideration? For papers that have mainly an empirical flavor it is typically good practice to show some benchmark studies. Burger's Equation is often considered as a standard problem, hence it should not be too difficult to provide some baselines for this problem.

* In Table 2 and 3, the difference between the proposed model and the ablation models appear to be marginal in terms of the RMSE. For example, is there a practical difference between 1.0e-3 and 1.2e-3 for the problems under consideration?

* Why does PINN h and h+H start to fail if the number of parameters are increased? Do you regularize these models?

* How did you chose the tuning parameters for the different models? I played around with the provided notebook "MPINN_2D" for a few minutes and I was able to improve the performance of the standard PINN quite a bit by slightly increasing the learning rate (2e-3), introducing a small amount of weight decay (5e-3) and switching off the learning rate scheduler. The PINN_15 achieves now an RMSE of 0.092 as compared to 0.14 by the proposed multilevel PINN. I am pretty sure that the performance can be further improved with some more careful fine-tuning of the hyperparamters. Tuning models are crucial and this needs to be addressed.

* The term 'robustness' typically means something different in the ML community. I am not exactly sure what you mean by `increased robustness of MPINNs'.

**Summary Of The Paper:**

This paper proposes a new physics informed neural network (PINN) that is motivated by multigrid methods. The idea of the proposed multilevel PINN is to write the solution for a given problem as sum of two terms, where the first term models fine-scale structures and the second term models coarse-scale structures. This approach yields models that converge faster and reduce the approximation error as compared to standard PINNs. The performance is demonstrated on both 1D and 2D problems.

**Summary Of The Review:**

This paper introduces a new multilevel PINN for learning solutions of PDEs. The presented ideas are novel and of interest to the scientific machine learning community. One shortcoming of this paper is that the experiments appear to be too preliminary, i.e., no benchmarks are provided and the models are not carefully fine-tuned. This makes it difficult to judge the advantage of the proposed method as compared to other state-of-the-art PINNs. At this point I feel that the paper is slightly below the acceptance threshold, but I am willing to change my score if the authors can address my questions above.

---

> ### Author Response · Authors · 2021-11-22
> **Answer to reviewer 2**
>
> We are really grateful to all the referees for their comments which helped us a lot to revise our paper.
>
> > How do you construct  $z_H$ and $z_h$? How do you determine whether  is not expressive enough? Is this an automatic process or does this require manual tuning? How is the performance affected if  is varied from less coarse to more coarse?
>
>  $z_H$ is just a coarser grid than $z_h$, we just select a $N_h\leq N_H$ and build the corresponding grid accordingly, with equi-spaced points. This is now described at the beginning of section 5. The behaviour of MPINNs with varying $N_H$ is now studied in the appendix. In terms of total number of operations we already have an advantage compared to "full PINNs" by training the coarse PINN on the same number of points than the fine PINN, cf. Figure 1 (d),  but the course PINN can indeed be trained on less points. We insisted more on this aspect in the 2D case, as in that case, being the grid 2 dimensional, the number of points is much larger and the difference may be more significant.  We have added a study related to both the hyperparameters N_H and H in the appendix.
> This is of course not a comprehensive study, but we can see that decreasing the number of points the performance of the MPINN also decreases and if too few points are used the MPINN does not converge.
>
> >In Table 2 and 3, the difference between the proposed model and the ablation models appear to be marginal in terms of the RMSE. For example, is there a practical difference between 1.0e-3 and 1.2e-3 for the problems under consideration?
>
> No there is no difference, but usually MPINNs are able to provide the result in less iterations and therefore with less operations.
>
> > Why does PINN h and h+H start to fail if the number of parameters are increased? Do you regularize these models?
>
>  The failure is due to the choice of the LR, we deduce from this that MPINNs are less sensitive to the choice of the LR (which is optimized for the case h=50, H=25), indeed MPINNs still perform well for larger h, while PINNs require a new LR tuning. However, we report now also the corresponding result of the training with BFGS, which uses a line search strategy for the step length selection. The results are still good.
>
> > How did you chose the tuning parameters for the different models? I played around with the provided notebook "MPINN2D" for a few minutes and I was able to improve the performance of the standard PINN quite a bit by slightly increasing the learning rate (2e-3), introducing a small amount of weight decay (5e-3) and switching off the learning rate scheduler. The PINN15 achieves now an RMSE of 0.092 as compared to 0.14 by the proposed multilevel PINN. I am pretty sure that the performance can be further improved with some more careful fine-tuning of the hyperparamters. Tuning models are crucial and this needs to be addressed.
>
> In the tests we have chosen the same learning rate (LR) for the three PINNs and we adjusted it manually for each problem. This is now stated at the beginning of Section 5. We believe the enhanced behaviour of MPINNs does not depend on the choice of the LR, as when we use LBFGS the LR is chosen by the linesearch routine implemented in the pytorch LBFGS method. We agree that the learning rate we chose may not be the best possible one, but our tests highlight also that MPINNs are less sensible to the choice of the learning rate as compared to PINNs. Finally, we point out that the main advantage of MPINNs is not to provide a lower RMSE (which sometimes is also the case) but rather to achieve the same error level of classical PINNs with less operations, cf. Figure 1d, and Figure 3.
>
>
> > The term 'robustness' typically means something different in the ML community. I am not exactly sure what you mean by `increased robustness of MPINNs'.
>
>  We agree with the referee, the term ”robust” is not appropriate here, we will change this.

---

> > ### Comment · Reviewer_9RP9 · 2021-11-27
> > **A few more questions**
> >
> > Thanks for addressing my concerns, but your response raised a few more questions.
> >
> > * You are saying that the MPINNs is able to provide the result in less iterations and therefore with less operations. This statement is questionable, since the MPINN requires more function evaluations. It would be better to compare the performance/convergence in terms of the number of function evaluations (NFE). Also, can you comment on the number of floating point operations per iterations for the MPINN and the standard PINN.
> >
> > * Do I understand you correctly that you optimize the learning rate for MPINNs on the model using h=50, H=25? Then you use the same learning rate for training PINNs and for models of varying sizes?  I feel that you need to provide more detailed experiments and evaluate both models over a range of learning rates to show that one model is less sensitive to the particular choice of the learning rate. At this point your statement that MPINNs are less sensitive may or may not be valid. It is hard to tell based on the current experiments.
> >
> > * I feel that there should be at least one set of experiments for which all models are fine-tuned, i.e., do a careful grid search. It is somewhat strange to fix the learning rate, and use the same learning rate for all models, even if you goal is to show that your model requires less iterations. Accuracy is important and currently it looks like that the PINNs are more accurate if fine-tuned. Please convince me otherwise by showing that the MPINN can match the accuracy of a fine-tuned PINN.

---

> > > ### Author Response · Authors · 2021-11-29
> > > **answer to the few more questions**
> > >
> > > We thank again all the referees for the second round of feedbacks, they are much appreciated.
> > >
> > > > The number of operations that we report in the tables and in the plots is the total number of floating point operations needed to compute the matrix-vector products during the training, so for instance for a network with one hidden layer, nh neurons in the hidden layer, nx training points and trained for niter iterations it would be 3*nx*nh*niter. For the MPINN this number takes into account both the operations performed at fine level and at coarse level (so at fine iterations when the coarse model is used, we add 3*nxH*nH*niter_coarse to the operations count)  that is all the matrix vector products performed to get the final result, so we believe our metric is equivalent to the one the referee suggests. In this way we can also take into account the fact that the operations performed with the coarse network do not have the same cost as the ones performed with the fine one, and this would be hidden by just counting the number of function evaluations
> > >
> > > > We did not optimise the LR for MPINNs on the model using h=50, H=25, we chose a LR that was yielding good results for all the three models on this test (MPINN and PINNH for instance give exactly the same result). Then we keep this same LR for higher values of h and H. Unluckily we did not have enough time to fine tune the networks in all the runs, and we relayed on the fact that we still have good performance with BFGS, which automatically select the best learning rate, to be convinced that the benefits of the method are not due to a bad choice of the learning rate. We understand however the referee's point of view, we did not fully address this point just for luck of time. We run some additional tests for the case h=50, H=25, for which we did a grid-search for the parameters gamma and step_size in scheduler = StepLR(optimizer, step_size, gamma), the results can be found here: https://www.overleaf.com/read/cgkydxpdkqdr. MPINN reaches the same accuracy as PINN but in less iterations.

---

### Official Review · Reviewer_Ji4t · 2021-11-02

**Correctness:** 3
**Technical Novelty And Significance:** 2
**Empirical Novelty And Significance:** 2
**Recommendation:** 3
**Confidence:** 4

**Main Review:**

Multigrid training for PINNs could be potentially interesting. In particular, I agree with the motivation of the authors that the particular structure of MPINNs could make training easier by imposing a kind of inductive bias. That being said, the results in this paper are too preliminary to be accepted.

1-D Linear Elliptic equation:
- It seems like the two PINNs you use only differ by capacity of the network. I thought the main motivation for Multigrid was that the PINN with less capacity would be trained on fewer points?
- The equation is chosen so that you get two different modes in the solution. It would be interesting to vary $\alpha$ to see whether MPINN training helps as the difference in frequency between the modes increases. I feel like the main motivation for this toy problem could be the different modes. It would be interesting to see it for more difficult parameter settings rather than only $\alpha = 3$ or $5$.
- Why the distinction between Adam for the 1-layer networks and BFGS for the 4-layer networks? Since BFGS uses n^2 memory, it seems Adam would be more appropriate for larger networks.
- How were the learning rates for the different problems tuned? I am not entirely convinced that the difference in performance is not from using different learning rates / hyperparameters for the different problems.
- You mention that MPINNs are more robust, but I am not sure I agree with the definition of robustness. It seems you are defining a budget of training epochs, and if the method does not converge in time, it is considered not robust. This would just mean the method is "slower", but not necessarily "less robust".
- It seems you are reporting mean + std of the RMSE. Can you report median + IQR instead?

2-D Nonlinear equation:
- Could you give some intuition on what the equation represents and what the solution looks like?
- 400 points vs 484 points does not seem like much of a difference between "fine" and "coarse". How did you choose those numbers?
- Results do seem less convincing in this case.

Burger's equation:
- What optimizer did you use?
- It seems you are picking different networks for each PDE. Here you have 3 hidden layers. How did you choose these sizes?

**Summary Of The Paper:**

The authors adopt ideas from multigrid methods in solving PDEs to Physics Informed Neural Networks (PINNs). After giving a standard introduction to PINNs and Multigrid methods, they describe their approach.

They decompose the PDE solution as a sum of a coarse and a fine term, and train independent PINNs to learn each term. The coarse PINN is chosen to have smaller capacity than the fine PINN, and is also (potentially) trained on fewer datapoints on the domain. Training proceeds by alternating training between the two PINNs: $\nu_1$ epochs on the fine PINN, followed by $\nu_2$ epochs on the coarse PINN, and repeat.

The authors test their method on a linear non-homogeneous 1-D elliptic equation (trained with Adam or BFGS), a 2-D nonlinear equation, and 1-D Burger's equation. They compare a standard PINN vs a larger PINN vs their method (MPINN).

**Summary Of The Review:**

The framework is potentially interesting, but the experiments in the paper do not convince me. It would be helpful to see more difficult problems, as well as multilevel problems with n > 2. Furthermore, there are a lot of details on hyperparameter tuning that is missing, leaving the question open whether the stated performance improvements are due to the hyperparameters or the MPINN idea.

---

> ### Comment · Reviewer_Ji4t · 2021-11-29
> **Response to response**
>
> Thank you for responding to my review. Although I would like to emphasize that the method is interesting and could have potential, and I appreciate the updates you made to the paper, the current setup and experiments are still not convincing to me and I would like to stick with my decision of rejection. I believe the changes required are major enough that I cannot justify acceptance. But for a future resubmission, here are my comments:
>
> Median and IQR:
> - Thanks for updating the paper to include these. I noticed though that often times the IQR is larger than the median? As an example in Table 2. It seems like the results are wildly varying for all the networks across different initializations. It would be very helpful to include a visualization to understand how much the IQR is compared to the median, without having to look at the numbers carefully.
>
> Operation count vs RMSE:
> - It seems like in the answer to another reviewer, you acknowledged that the RMSE difference is not major enough, and that the focus should be operation count. In that case, you should include a detailed description of how operation count was defined, and include it in all of your tables of results. Also, since these are run on the same computer, would it be possible to compare real-time as well? That would give a much more direct comparison.
>
> Hyperparameter tuning:
> - Since it is known that NN performance is extremely dependent on hyperparameters, these should be carefully tuned (using grid-search, random-search, bayesopt) for each different architecture/training method. Specifically, I would have liked to see weight decay parameters and learning rate tuned. I don't think hand tuning is sufficient in this case unfortunately, especially when comparing training iterations. Learning rate + weight decay could have a huge influence on training iterations.

---

> > ### Author Response · Authors · 2021-11-29
> > **Response to reviewer's response**
> >
> > We thank again all the referees for the second round of feedbacks, they are much appreciated.
> >
> > >Median and IQR:
> > Thanks for the suggestion, we will include this in the revision.
> >
> > > Operation count vs RMSE:
> > Thanks for the suggestion, we will include this in the revision. The computational time is not directly comparable, because the training of PINNs is optimized in pytorch, while the multilevel mechanism is not, and the passage from fine to coarse and vice-versa slows the training down in terms of computational time.
> >
> > > Hyperparameter tuning:
> > Unluckily we did not have enough time to fine tune the networks in all the runs, and we relayed on the fact that we still have good performance with BFGS, which automatically select the best learning rate, so we are confident that the benefits of the method are not due to a bad choice of the learning rate. As the referee pointed out, we wanted to emphasize the gain on the operations counter and that's why we report many results with GD. Indeed the operations count is difficult to do with BFGS, while it is straightforward with GD, which has a much simpler implementation. We understand however the referee's point of view, we did not fully address this point just for luck of time.

---

> > > ### Author Response · Authors · 2021-11-29
> > > **Few additional runs with grid-search**
> > >
> > > We run some additional tests for the case h=50, H=25, for which we did a grid-search for the parameters gamma and step_size in scheduler = StepLR(optimizer, step_size, gamma), the results can be found here: https://www.overleaf.com/read/cgkydxpdkqdr. MPINN reaches the same accuracy as PINN but in less iterations.

---

### Author Response · Authors · 2021-11-22
**Answer to reviewer 1**

We are really grateful to all the referees for their comments which helped us a lot to revise our paper.

> It seems like the two PINNs you use only differ by capacity of the network. I thought the main motivation for Multigrid was that the PINN with less capacity would be trained on fewer points?
In terms of total number of operations we already have an advantage compared to "full PINNs" by training the coarse PINN on the same number of points than the fine one, cf. Figure 1 (d),  but the coarse PINN can indeed be trained on less points. We have added a study related to the hyperparameters tuning for MPINNs in the Appendix (runs for $N_H<N_h$ and $H\neq h/2$).
This is of course not a comprehensive study, but we can see that decreasing the number of points the performance of the MPINN also decreases and if too few points are used the MPINN does not converge

> The equation is chosen so that you get two different modes in the solution. It would be interesting to vary  to see whether MPINN training helps as the difference in frequency between the modes increases.

We have added this study in Table 2.

> Why the distinction between Adam for the 1-layer networks and BFGS for the 4-layer networks? Since BFGS uses $n^2$ memory, it seems Adam would be more appropriate for larger networks.


 The aim of this test was just to show that the proposed procedure works with any kind of optimizer, and not just with gradient methods. We could as well have used the same network that we use with Adam, but we used a deeper network because BFGS is faster than Adam.    We actually use the LBFGS implementation in pytorch, so the memory storage is far less than $n^2$. This is now stated in the text


> How were the learning rates for the different problems tuned? I am not entirely convinced that the difference in performance is not from using different learning rates / hyperparameters for the different problems.

 In the tests we have chosen the same learning rate (LR) for the three PINNs and we adjusted it manually for each problem. This is now stated at the beginning of Section 5. We believe the enhanced behaviour of MPINNs does not depend on the choice of the LR, as we report also runs performed with LBFGS in which the LR is chosen by the linesearch routine implemented in the pytorch LBFGS method. We agree that the learning rate we chose may not be the best possible one, but our tests highlight also that MPINNs are less sensitive  to the choice of the learning rate as compared to PINNs, for instance when we increase the size of the network, cf. Table 1, MPINN still performs well with the chosen LR, while PINNh requires a new tuning, as its limited performance is actually caused by the optimizer, as shown by   the use of BFGS on the same problems, cf. Figure 2

> You mention that MPINNs are more robust, but I am not sure I agree with the definition of robustness. It seems you are defining a budget of training epochs, and if the method does not converge in time, it is considered not robust. This would just mean the method is "slower", but not necessarily "less robust".

We agree with the referee, the term "robust" is not appropriate here, we will change this.

> It seems you are reporting mean + std of the RMSE. Can you report median + IQR instead?
 Done!

2-D Nonlinear equation:
> Could you give some intuition on what the equation represents and what the solution looks like?
 This is a nonlinear equation often used when testing optimization methods on nonlinear PDEs, we added a plot of the solution in the revision.

> 400 points vs 484 points does not seem like much of a difference between "fine" and "coarse". How did you choose those numbers?
Results do seem less convincing in this case.

Additional runs have been added in the Appendix
>Burger's equation:  What optimizer did you use?
It seems you are picking different networks for each PDE. Here you have 3 hidden layers. How did you choose these sizes?

We used Adam optimizer. The networks architectures were hand-tuned, and for each problem we picked the "less complex one" that yields a solution approximation that "visually" approximates well the ground-truth.

> Summary Of The Review:
We agree with the referee that the paper could benefit from the addition of tests with $n>2$ levels. In our experience if a multilevel method works, it already works in the two level setting and the addition of more levels increases the performance. We leave this study as a meaningful future research direction, together with the theoretical study of the approach. As for the LR, as we motivated above, maybe the tuning of the learning rate is not the best possible one, but we don't think that the results depend on this choice. The LR is the same for all networks and it was not chosen to favour any of them.  In particular with BFGS we use linesearch and not a hand-tuned strategy and the results are good.

---

### Decision · Program_Chairs · 2022-01-20

**Decision:**

Reject

**Comment:**

The paper develops an instance of physics informed neural network inspired from multigrid methods for solving PDEs. The proposed framework describes the solution of a PDE problem as the sum of terms operating at different resolutions. Training is performed by an iterative optimization algorithm that alternates between the different resolution models. Experiments are performed on 1D and 2D problems.

All the reviewers agree on the originality and the potential of the proposed method. They however all consider that the current version of the work is too preliminary both in the form and in the content. The experimental contribution should be developed further with tests performed on more complex problems and complementary analyses. Some of the claims should be given more evidence or moderated. It also appeared during the discussion that the models are not well tuned, making the results inconclusive. The authors are encouraged to develop and strengthen their work.